# Development and Validation of Deep Learning-Based Algorithms for Predicting Lumbar Herniated Nucleus Pulposus Using Lumbar X-rays

**DOI:** 10.3390/jpm12050767

**Published:** 2022-05-09

**Authors:** Jong-Ho Kim, So-Eun Lee, Hee-Sun Jung, Bo-Seok Shim, Jong-Uk Hou, Young-Suk Kwon

**Affiliations:** 1Department of Anaesthesiology and Pain Medicine, Chuncheon Sacred Heart Hospital, College of Medicine, Hallym University, Chuncheon 24253, Korea; poik99@hallym.or.kr; 2Division of Big Data and Artificial Intelligence, Institute of New Frontier Research, Chuncheon Sacred Heart Hospital, College of Medicine, Hallym University, Chuncheon 24253, Korea; 3Division of Software, Hallym University, Chuncheon 24252, Korea; dlth508@naver.com (S.-E.L.); glee623@naver.com (H.-S.J.); bycicle55@naver.com (B.-S.S.)

**Keywords:** herniated nucleus pulposus, lumbar X-ray, deep learning, prediction

## Abstract

Lumbar herniated nucleus pulposus (HNP) is difficult to diagnose using lumbar radiography. HNP is typically diagnosed using magnetic resonance imaging (MRI). This study developed and validated an artificial intelligence model that predicts lumbar HNP using lumbar radiography. A total of 180,271 lumbar radiographs were obtained from 34,661 patients in the form of lumbar X-ray and MRI images, which were matched together and labeled accordingly. The data were divided into a training set (31,149 patients and 162,257 images) and a test set (3512 patients and 18,014 images). Training data were used for learning using the EfficientNet-B5 model and four-fold cross-validation. The area under the curve (AUC) of the receiver operating characteristic (ROC) for the prediction of lumbar HNP was 0.73. The AUC of the ROC for predicting lumbar HNP in L (lumbar) 1-2, L2-3, L3-4, L4-5, and L5-S (sacrum)1 levels were 0.68, 0.68, 0.63, 0.67, and 0.72, respectively. Finally, an HNP prediction model was developed, although it requires further improvements.

## 1. Introduction

Lumbar disk herniation is defined as the localized displacement of the disk material beyond normal margins of the intervertebral disk space [1]. The incidence of disk herniation is approximately 5–20/1000 adults annually, and it is most common among adults in their third–fifth decades, with a male to female ratio of 2:1 [2]. Disk disease is an underlying cause of lower back pain in fewer than 5% of patients [3]. A physical examination focused on the lumbar 4-5 or lumbar 5-sacral 1 level is recommended [4], as lumbar herniated disks almost develop at the lumbar 4-5 or lumbar 5-sacral 1 level.

The purpose of a physical examination is to identify features that suggest that further evaluation is necessary, rather than arriving at a primary diagnosis. The first clue to lumbar disk herniation is sciatica, which can be caused by other disorders. Therefore, the image diagnosis test can be helpful in diagnosing lumbar herniated disks. Plain lumbar radiography is widely performed in patients with lumbar pain. It is easily accessible in both clinics and outpatient clinics. This imaging technique can be used to evaluate structural instability [5] and is helpful to rule out tumors, fractures, infections, and spondylolisthesis; however, it does not show a herniated disk [6]. Although high-quality studies are insufficient for advanced imaging in patients with lumbar disk herniation, magnetic resonance imaging (MRI) is recommended to confirm the presence of lumbar disk herniation. It is the most appropriate noninvasive test in patients with a history and physical examination suitable for lumbar disk herniation [1]. MRI is the preferred and sensitive method for visualizing disk herniation. MRI findings are helpful when a surgeon and other clinicians are establishing a plan for the procedure [5]. However, MRI examination is expensive and is not suitable for some patients with MRI contraindications. If a clinician is aware of the presence of a herniated lumbar disk at each lumbar level with simple and low-cost tests (e.g., plain lumbar radiography), it would be helpful in the diagnosis and treatment of patients with low back pain or sciatica. 

This study aims to develop an artificial intelligence model that predicts the existence of a herniated lumbar disk at each lumbar level from a plain lumbar X-ray image. Furthermore, the study establishes the detection of lumbar disk herniation in a lumbar X-ray by using the artificial intelligence model and gradient-weighted class activation mapping (Grad-CAM).

## 2. Materials and Methods

### 2.1. Data Source and Study Population

In this retrospective study, data were collected from a cohort enrolled at four hospitals in Hallym University Medical Center, Republic of Korea. The study assumed that adult patients (i.e., ≥18 years old) have lumbar images of both MRI and plain X-ray, from January 2010 to February 2021. The study excluded patients with no MRI readings and those who underwent surgeries associated with the lumbar disk between the date of X-ray examination and the date of MRI examination. The study protocol was approved by the Institutional Review Board of Chuncheon Sacred Hospital (CHUNCHEON 12 March 2020), which waived the requirement for informed consent. All procedures in this study were performed according to relevant guidelines and regulations.

### 2.2. Herniated Lumbar Disk and Lumbar X-ray

During the course of the study, HNP was determined using MRI readings. The interval between the X-ray and MRI test was determined within 3 months. Labeling of the presence or absence of HNP was performed on all patients included in this study according to MRI readings. Labeling was performed at each level of the lumbar spine: L (lumbar) 1-2, L2-3, L3-4, L4-5, and L5-S (sacrum)1. All X-ray images were collected from the picture archiving and communication system (PACS) in Joint Photographic Experts Group (JPEG) format. The lumbar X-ray images included anterior–posterior, lateral, bilateral oblique, flexion, and extension views. There were different numbers of images, from one to six, according to the patients, and the data were collected two times. The first set of data was collected from January 2010 to March 2020, and the second set was collected from March 2020 to February. The initial data were used to train the model, with the second being used to test the last model.

### 2.3. Deep Learning Model and Training Strategy

The study uses the EfficientNet-B5 model [7] as the base convolutional neural networks (CNN), which is widely used for image classification tasks. It achieves a higher accuracy and efficiency level than previous CNN backbone networks. The study uses a pretrained network based on the ImageNet database and applies a learning rate of 3 × 10^−5^ with cosine annealing with a warm restart scheduler and Adam optimization.

Data augmentation is used to improve the classification performance of the CNN model. It helps prevent overfitting and improves the generalization performance of the neural network model. The study uses image resizing and contrast-limited adaptive histogram equalization [8], a method of adjusting the image contrast based on spreading the image histogram at both ends when image pixels are distributed around a certain range of values. To emphasize the region of interest in the X-ray images, the contrast of pixel intensity between the dark and bright areas is increased to make the disk area more distinctive. Note that augmentations with image flip and rotation are excluded because the model may confuse disks L1-2 to L5-S1 in a reversed image.

The training dataset consisted of 162,257 lumbar X-ray images from 31,149 patients, and model training was controlled based on four-fold cross-validation. Images with the same patient ID were assigned to the same fold to prevent data leakage between the validation and training datasets. Training was performed on an Nvidia Geforce RTX 3090 and took approximately 50 min per epoch. The total training time was approximately 2 days and 18 h. The score was calculated on the basis of the optimal threshold for each fold. The performance of the model was evaluated using precision, recall (sensitivity), f1-score, and area under the curve (AUC) score of the receiver operating characteristic (ROC) curve. Performance was also evaluated through the AUC of the ROC curve and the f1-score, which is the harmonic average of precision and recall because of the unbalanced training dataset for each herniated lumbar disk. 

### 2.4. Deep Learning Model Testing and Evaluation

The test dataset consisted of 18,014 lumbar X-ray images obtained from 3512 patients. A more accurate prediction was derived by combining predictions using a bagging ensemble [9], with four weights obtained through four-fold training data. During the validation, prediction was obtained using the optimal threshold for each data fold. Confidence probability scores from all images were evaluated, and the average value of the predicted probability for each patient was obtained for the final decision. In addition, Grad-CAM [10] was applied by sorting the data in descending order based on the confidence value and storing the top 300 images whose disks were identified with a high probability. The study selected the last convolution layer for Grad-CAM visualization.

## 3. Results

During the second data collection period, the data from 3478 patients were included in this study. However, those from 92 patients were excluded since the MRI reading consisted of other parts away from the lumbar area. A total of 3037 patients were excluded owing to missing data. The training dataset used in each experiment was the same as that of the 162,257 lumbar X-ray images from 31,149 patients. The test dataset included 18,014 lumbar X-ray images of 3512 patients. The study attempted various evaluation methods to improve overall performance. The distribution of each lumbar HNP level according to each fold in the training set is summarized in Table 1.

### 3.1. Baseline Task

For the baseline task, an experiment was conducted to determine whether a herniated lumbar disk was present. If there is at least one disk for each level in a single image, it was determined that there is a disk.

For the baseline task, we conducted an experiment to determine whether there was a herniated lumbar disk through four-fold cross-validation. If there is at least one disk for each level in a single image, it was determined that there is a disk. The results of the HNP prediction are summarized in Table 2.

### 3.2. Herniated Disk Diskrimination Using Lumbar Level

Since the ultimate goal was to identify the level at which the disk was located, an experiment was also conducted to determine the presence of a disk for each level through four-fold cross-validation. For each lumbar level, training was performed to determine the presence of a disk using binary cross-entropy with logit loss. The results of the prediction of the presence of HNP at each level of the lumbar spine are summarized in Table 3.

### 3.3. Multitask Learning

Multitask learning is a technique used to improve the overall performance of all tasks by simultaneously learning related tasks. Multitask learning with image data and age information was conducted to enhance optimization performance through four-fold cross-validation since disks occur more frequently in relatively elderly people. The results of the multitasking are summarized in Table 4.

### 3.4. Deep Learning Model Evaluation Using Test Set

The evaluation of the final deep learning model using the test set for the prediction of HNP presence at each lumbar level through baseline task and multitask is summarized in Table 5.

### 3.5. Grad-CAM

In Grad-CAM, some showed heat maps in the lumbar disk area and some showed heat maps along the lumbar vertebrae shape; however, some showed heat maps in areas not associated with HNP (Figure 1).

## 4. Discussion

The study developed an artificial intelligence to predict lumbar HNP using lumbar radiography. To train the predictive lumbar HNP model, data from 31,149 patients were used. Data from 3512 patients were used to evaluate the developed model. The EfficientNet-B5 model achieved an AUROC of 0.63–0.72 according to lumbar level, but it is still insufficient for acceptance as a diagnosis tool. However, this study suggests the possibility of detecting lumbar HNP utilizing lumbar X-ray using CNN. Grad-CAM showed that the model detects HNP with vertebrae and intervertebral disk or vertebral alignment, although there were cases of heatmap unrelated with HNP.

X-ray examination is a fundamental diagnostic modality for patients with low back pain. Intervertebral space narrowing, traction osteophytes, and compensatory scoliosis on X-rays are commonly suggestive of HNP [9]. However, it is difficult to detect herniated disks on plain radiographs since they present low value in the diagnosis of lumbar disk degeneration and herniation [10]. Clinically, lumbar X-rays are not only used for HNP diagnosis. Acute HNP can be better diagnosed on good-quality MR than myelography, diskography, and postmyelography/postdiskography computed tomography because they are ordinarily reserved for equivocal and protracted cases. MRI has a diagnosis accuracy of 97% and is the most sensitive examination to visualize HNP, with significant ability for soft tissue visualization. It also has a higher interobserver reliability than other imaging examinations. MRI is recommended as the initial method of choice for the diagnosis of HNP [11]. However, MRI scans are expensive and difficult to use as screening tests and can be used when patient outcomes are worse due to delayed treatment [12]. Therefore, cost-effective and accurate examination is required to detect HNP. 

The application of AI models in medicine is gradually increasing, which includes AI models as medical tools in the spine area. Several studies have used deep learning with lumbar X-ray imaging. Zhang et al. reported that a deep learning model has the potential to classify osteopenia and osteoporosis [13]. Mbaki et al. reported that the lumbar spine disk classification method is based on deep convolutional neural networks using axial-view MRI [14]. They also reported using a method based on deep learning for herniated lumbar disk segmentation [15]. Tsai et al. reported a method based on deep learning for the automatic detection of lumbar disk herniation in MRI [16]. Pan et al. reported that an automatic diagnosis system using deep learning can classify images of normal disks, disk bulges, and disk herniation [17]. Previous AI-based studies on the spine were not classifications that humans could difficultly detect outcome in images. Although imperfectly, humans can detect it to some extent. No study has used lumbar radiographs for the detection or classification of HNP, which is a common spinal disease. Till this date, studies on the classification of HNP have used MRI. It is difficult to find HNP by lumbar X-rays themselves and some cases may remain unreported due to poor results. In addition, transverse images are essential for diagnosing HNP using MRI [11]. However, lumbar X-ray images do not include transverse images. Therefore, a predictive model of HNP using lumbar X-rays cannot detect HNP directly in lumbar X-rays, and it is likely to detect HNP using other characterized patterns.

The principle of X-ray imaging is that the rays generated in an X-ray tube penetrate the human body and are attenuated owing to the type and density of substances in the body. Various medically helpful diagnoses were made using this image after imaging this weakened part using a detector. The difference with photography is that photographs detect an image’s light reflected from an object, while X-ray tests detect an image’s radiation transmitted through the object. X-rays do not penetrate the bone but penetrate the water or blood [18]. Therefore, the main field of application of X-ray imaging is the examination of fractures and changes in the skeletal system. Since the intervertebral disk consists of annulus fibrosus, a dense collagen ring of annulus fibrosus that surrounds the nucleus pulposus, it may be difficult to find the intervertebral disk with the naked eye in X-ray images. However, some soft tissue is visible in the X-ray images with the naked eye. We expected that an AI prediction model could find lumbar HNP with the findings not visible through naked eyes in X-ray images but through differences in images.

Although our model did not easily detect lumbar HNP from lumbar X-rays results, this study aimed to suspect lumbar HNP from lumbar X-rays using Grad-CAM. In the Grad-CAM results, the correct prediction of patients with HNP varied. Some (Figure 1) showed a heat map around the disk and vertebrae, where the actual HNP was present in the anteroposterior view of the lumbar radiograph. HNP is a consequence of degenerative changes in the annulus. Although the disk itself does not appear on the X-ray, the changes related to degeneration such as osteophytes and subchondral sclerosis are visible on an X-ray. Osteophyte severity and endplate sclerosis has a stronger association with lumbar intervertebral disk degeneration [19]. The predicting model could detect these changes. Some results show a heat map around the disk where the actual HNP was present and along the vertebrae line in the lateral view of the lumbar X-ray image. These heat maps show that the disk and surrounding structures may be important for the prediction model of lumbar HNP based on deep learning. Abnormal spine alignment can be associated with HNP in L4-5 [20]. In Figure 1c, the heat map was stronger in the facet joint of L4-5 and L5-S1. Thus, the model can classify based on the status of facet joints. Lumbar X-ray can also show pathological changes, especially in severe diseases characterized by articular space stenosis, sclerosis, subchondral sclerosis and erosion, cartilage thinning, calcification of the articular sac, and enlargement of the articular process [21]. According to a recent study on detecting lumbar spinal stenosis using AI, hyperdense facet joint in lumbar X-ray may be a sign of lumbar spinal stenosis [22]. Although we can check the degree of alignment using Grad-CAM images to some extent, facet joint abnormalities need to be checked with original images or MRI images of patients with similar pattern Grad-CAM images because it is difficult to check facet joint abnormalities using Grad-CAM.

This study identified some limitations. First, an AI model was developed that predicted the presence of HNP at each lumbar level; however, the model did not exhibit excellent performance. One of the reasons for the inaccurate model performance was the unbalanced dataset. Approximately 80,000 from the entire training dataset consisted of patient data without a disk at all. The number of data points for each disk level was also unbalanced. On average, more disks occurred in L5-S1 than in L1-2; this trend is also reflected in the image-label distribution. Therefore, collecting more data from patients with insufficient disk numbers is a possible solution. Because the texture of the disk does not appear on the X-ray image, it cannot be distinguished, even by an experienced radiologist. In addition, the X-ray image is the result of two-dimensional projection. Therefore, there is a clear limitation in diagnosing the three-dimensional structure from the X-ray image. Therefore, depending on the direction, it may not be possible to identify a herniated disk. X-ray images obtained from multiple directions could be a solution to this problem. Images from our dataset consist of anterior–posterior, lateral, left oblique, right oblique, flexion, and extension views for each patient. Considering this data distribution, the study plans to improve the performance of the model through indirect 3D structure identification using multiview classification [23] in the future. Second, manipulation of images or checking for other spine pathologies may be needed to detect HNP using Grad-CAM, as there were cases with broad or unrelated heat maps. Therefore, training is required after removing unrelated parts of the spine in the images, and other possible pathologies must be investigated. This process can lead to improved performance. Third, the severity and direction (left, central, and right) of HNP are also important, but the prediction model of HNP only predicts the presence of HNP. In the study, the model predicted each level of the lumbar HNP. If the severity and direction of the HNP are added to the target of prediction, the targets become excessive, meaning the data were insufficient for each target. The present prediction model does not have positive target data for HNP of the upper lumbar level. If a new model has more targets than the present one, it may be difficult to predict the severity and direction of HNP. A recent study reported diagnostic triage in patients with central lumbar spinal stenosis using a deep learning system of radiographs. To improve the performance of our model, their proposed framework may be considered [22].

## 5. Conclusions

This study developed a model to predict HNP based only on lumbar radiographs and showed potential usefulness for detecting HNP using lumbar X-ray. Extensive experiments were performed using real clinical datasets to evaluate the applicability of the model to routine clinical practice. To the best of our knowledge, this is the first peer-reviewed report on a model for predicting HNP using lumbar X-ray images. To improve model performance, other technical methods should be considered; we investigated association with other pathologic findings.

## Figures and Tables

**Figure 1 jpm-12-00767-f001:**
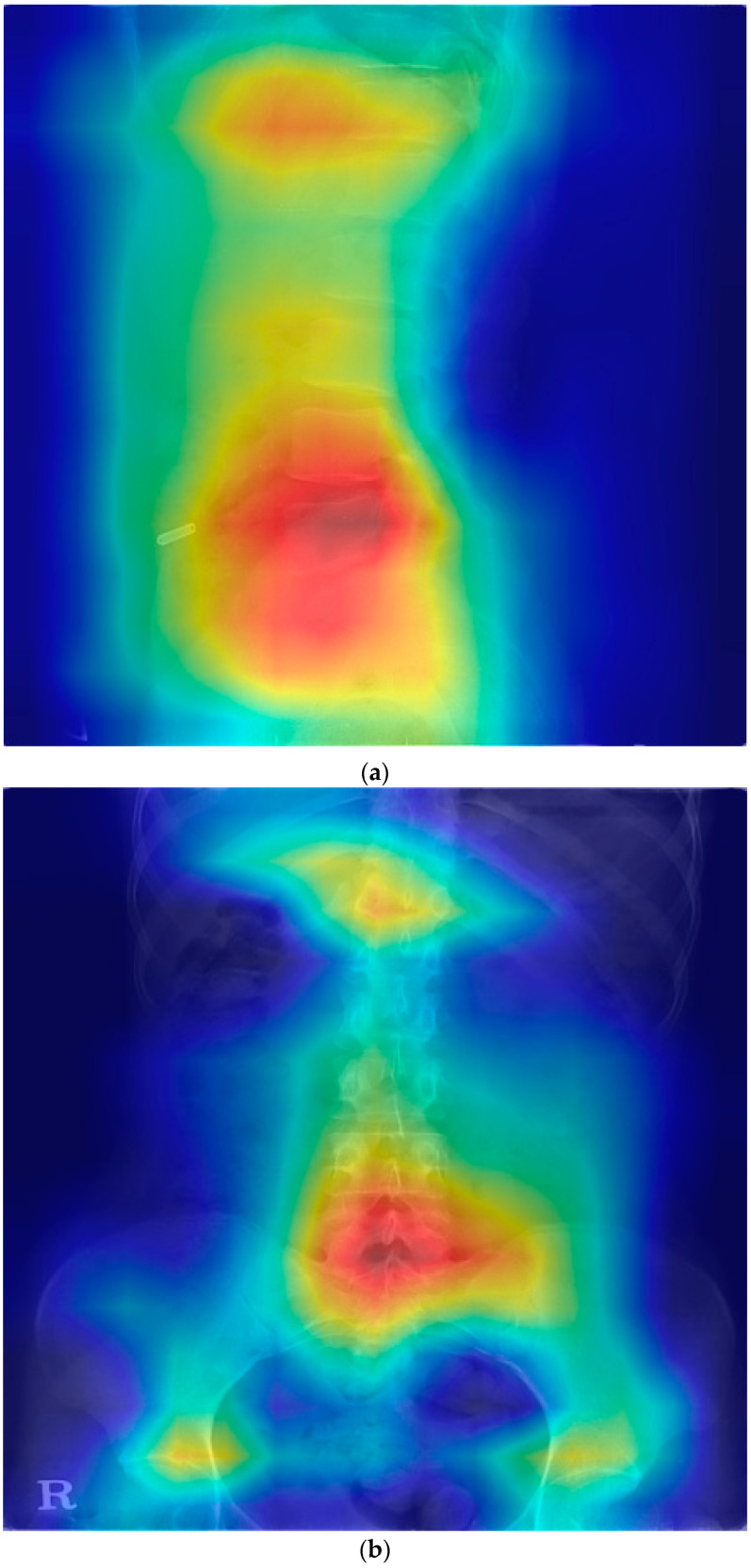
Gradient-weighted class activation mapping for the prediction model of lumbar HNP. HNP, herniated nucleus pulposus; L, lumbar; S, sacrum. (**a**) Heatmap in lateral view of the disk and vertebrae area (HNP in L4-5 and L5-S1). (**b**) Heatmap in anteroposterior view of the disk and vertebrae area (HNP in L5-S1). (**c**) Heat map along the posterior vertebral line (HNP in L4-5 and L5-S1). (**d**) Heap map in the area unrelated with HNP (HNP in L4-5 and L5-S1).

**Table 1 jpm-12-00767-t001:** Each lumbar level distribution of lumbar herniated nucleus pulposus according to each fold in the training set.

		L1-2	L2-3	L3-4	L4-5	L5-S1
Fold 0	No-HNP	39,331	38,166	36,058	28,765	30,559
	HNP	1360	2525	4633	11,926	10,132
	Sum	40,691	40,691	40,691	40,691	40,691
Fold 1	No-HNP	39,306	38,174	35,929	28,981	30,532
	HNP	1282	2414	4659	11,607	10,056
	Sum	40,558	40,558	40,558	40,558	40,558
Fold 2	No-HNP	39,108	37,779	35,902	28,619	30,393
	HNP	1263	2592	4469	11,752	9978
	Sum	40,371	40,371	40,371	40,371	40,371
Fold 3	No-HNP	39,471	38,244	36,201	28,578	30,411
	HNP	1135	2362	4405	12,028	10,195
	Sum	40,606	40,606	40,606	40,606	40,606

L, lumbar; S, sacrum; HNP, herniated nucleus pulposus.

**Table 2 jpm-12-00767-t002:** Precision, recall, f-1 score, accuracy, and ROC AUC for HNP presence prediction.

		Fold 0	Fold 1	Fold 2	Fold 3
	Precision	0.67	0.68	0.66	0.65
No-HNP	Recall	0.66	0.69	0.67	0.7
	F1-score	0.67	0.68	0.66	0.67
	Precision	0.67	0.68	0.68	0.68
HNP	Recall	0.68	0.67	0.66	0.63
	F1-score	0.67	0.67	0.67	0.66
	Accuracy	62.3%	66.1%	58.8%	65.2%
	AUC	0.73	0.74	0.71	0.73

HNP, herniated nucleus pulposus; AUC, area under curve.

**Table 3 jpm-12-00767-t003:** Results for predicting the presence of HNP in each level of lumbar through four-fold cross-validation.

		Fold 0	Fold 1	Fold 2	Fold 3
L1-2	Precision	0.06	0.07	0.06	0.05
	Recall	0.7	0.64	0.72	0.73
	F1-score	0.11	0.12	0.12	0.1
	AUC	0.69	0.73	0.74	0.74
L2-3	Precision	0.11	0.12	0.13	0.12
	Recall	0.62	0.64	0.58	0.65
	F1-score	0.19	0.2	0.12	0.2
	AUC	0.7	0.71	0.69	0.73
L3-4	Precision	0.2	0.2	0.19	0.19
	Recall	0.56	0.57	0.62	0.57
	F1-score	0.3	0.29	0.29	0.29
	AUC	0.68	0.68	0.69	0.68
L4-5	precision	0.43	0.41	0.41	0.44
	Recall	0.68	0.71	0.67	0.66
	F1-score	0.52	0.52	0.51	0.53
	AUC	0.7	0.7	0.68	0.71
L5-S1	Precision	0.4	0.39	0.39	0.41
	Recall	0.68	0.73	0.64	0.64
	F1-score	0.51	0.5	0.48	0.5
	AUC	0.73	0.72	0.7	0.72
	Accuracy	65.5%	67.2%	66.7%	67.0%

L, lumbar; S, sacrum; AUC, area under curve.

**Table 4 jpm-12-00767-t004:** Multitask results for predicting the presence of HNP in each level of lumbar through four-fold cross-validation.

		Fold 0	Fold 1	Fold 2	Fold 3
L1-2	Precision	0.05	0.05	0.06	0.05
	Recall	0.66	0.66	0.64	0.71
	F1-score	0.09	0.1	0.1	0.1
	AUC	0.64	0.67	0.68	0.69
L2-3	Precision	0.09	0.09	0.11	0.08
	Recall	0.74	0.74	0.54	0.77
	F1-score	0.16	0.15	0.19	0.15
	AUC	0.66	0.65	0.66	0.66
L3-4	Precision	0.18	0.16	0.16	0.17
	Recall	0.55	0.59	0.62	0.51
	F1-score	0.28	0.25	0.26	0.26
	AUC	0.66	0.63	0.65	0.64
L4-5	Presicion	0.42	0.4	0.42	0.4
	Recall	0.68	0.71	0.66	0.74
	F1-score	0.52	0.51	0.51	0.52
	AUC	0.7	0.69	0.69	0.69
L5-S1	Precision	0.4	0.39	0.39	0.37
	Recall	0.63	0.73	0.67	0.65
	F1-score	0.49	0.5	0.49	0.47
	AUC	0.71	0.72	0.7	0.69
	Accuracy	61.5%	60.1%	65.0%	60.5%

L, lumbar; S, sacrum; AUC, area under curve.

**Table 5 jpm-12-00767-t005:** Evaluation results of the final deep learning model using the test set for predicting HNP presence in each lumbar level.

	Precision	Recall	F1-Score	ROC AUC	Accuracy
L1-2	0.05	0.6	0.09	0.68	69.1%
L2-3	0.09	0.65	0.15	0.68	
L3-4	0.18	0.38	0.24	0.63	
L4-5	0.41	0.47	0.43	0.67	
L5-S1	0.36	0.55	0.43	0.72	

HNP, herniated nucleus pulposus; ROC, receiver operating characteristic; AUC, area under the curve.

## Data Availability

All data were obtained from Hallym University Medical Center. Data can be used after obtaining permission from Hallym University Medical Center.

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
