# Peer review of "Development and Validation of Deep Learning-Based Algorithms for Predicting Lumbar Herniated Nucleus Pulposus Using Lumbar X-rays"

_jpm, 2022, doi:10.3390/jpm12050767_

Round 1
Reviewer 1 Report
This study applied a deep learning model for an image classification task, i.e., determining whether herniated nucleus pulposus (HNP) presents on X-ray radiographers. I can not understand the rationale of the study. It is well-known that X-rays can not detect HNP, and deep learning can not “magically” generate information that was not collected.
Author Response
Reviewer 1.
Comment
This study applied a deep learning model for an image classification task, i.e., determining whether herniated nucleus pulposus (HNP) presents on X-ray radiographers. I can not understand the rationale of the study. It is well-known that X-rays can not detect HNP, and deep learning can not “magically” generate information that was not collected.
Answer
Thank you for the valuable comment. As you mentioned, it is very difficult to detect HNP using X-ray. Although the motive of this study was to detect HNP using deep learning, it was also the main purpose of this study to discover findings suspicious of HNP using Grad-CAM. We corrected and described the purpose of this study in Introduction , and the results of Grad-CAM were analyzed based on the existing literature related to HNP findings in the discussion.
line 56-58
line 236-258
Reviewer 2 Report
The authors present a Deep Learning-Based Algorithms for Predicting Lumbar Herniated Nucleus Pulposus Using Lumbar x-rays. Despite the topic may be interesting for the readers and may contribute to futher research in the field of spine research, I kindly recommend the authors to reanalyze some aspects of the manuscript:
- I suggest the authors to reprhase the introduction and the discussion sections. For example, the discussion section starts with a long paragraph that belongs to results section (line 185-193). Moreover, the discussion section is weak: the authors do not place their results into the literature knowledge. Most of the part of the discussion section appreas to be in fact an introduction (see lines 194-229). I suggest the authors to better explain/integrate their results into the existing knowledge, as a part of it not as an independent fact.
- despite the Grad Cam visualisation was used in this experiment, the authors fail to interpret (and even to present) the obtained results. I suggest the authors to reanalize the GradCam discussion section acordingly (line 230-239).
- Based on the aboved mentioned, I consider the discussion section as a major gap of the manuscript. Furthermore, the introduction section, but also the Conclusion one should be reanalyzed.
Author Response
Reviewer 2.
Comment
The authors present a Deep Learning-Based Algorithms for Predicting Lumbar Herniated Nucleus Pulposus Using Lumbar x-rays. Despite the topic may be interesting for the readers and may contribute to futher research in the field of spine research, I kindly recommend the authors to reanalyze some aspects of the manuscript:
I suggest the authors to reprhase the introduction and the discussion sections. For example, the discussion section starts with a long paragraph that belongs to results section (line 185-193). Moreover, the discussion section is weak: the authors do not place their results into the literature knowledge. Most of the part of the discussion section appreas to be in fact an introduction (see lines 194-229). I suggest the authors to better explain/integrate their results into the existing knowledge, as a part of it not as an independent fact.
despite the Grad Cam visualisation was used in this experiment, the authors fail to interpret (and even to present) the obtained results. I suggest the authors to reanalize the GradCam discussion section acordingly (line 230-239).
Based on the aboved mentioned, I consider the discussion section as a major gap of the manuscript. Furthermore, the introduction section, but also the Conclusion one should be reanalyzed.
Answer
Thank you for the detailed comment and suggestion. We revised the manuscript according to your suggestions. The revisions are highlighted in yellow in the revised manuscript at the following locations.
line 56-58
line 184-189
line 201-203
line 216-222
line 236-258
line 275-288
line 291
line 294-296
Reviewer 3 Report
Reading the manuscript: Development and validation of deep learning-based algorithms for predicting lumbar using lumbar x-rays written by Kim et al., was really interesting. The authors’ results highlighted an HNP prediction model, but need improvement with further studies on this topic.
There are however some points to consider which I think will improve the understanding and coherence of this manuscript:
- Please correct the typos errors. This will apply to the whole manuscript. Other typos include lack of space before/ after E.g. lines: 23-24: in L(lumbar)1-2, L2-3, L3-4, L4-5, and L5-S(sacrum)1; 74-75: L(lumbar) 1-2, L2-3, L3-4, L4-5, L5-S(sacrum)1; 84: EfficientNet-B5 model[7]; 257: Second, The severity.
- It would be interesting to correlate the data in the sections. The information in the introduction and discussion sections needs to be enriched with more recent data and more appropriately correlated.
Author Response
Reviewer 3.
Comment
Reading the manuscript: Development and validation of deep learning-based algorithms for predicting lumbar using lumbar x-rays written by Kim et al., was really interesting. The authors’ results highlighted an HNP prediction model, but need improvement with further studies on this topic.
There are however some points to consider which I think will improve the understanding and coherence of this manuscript:
Please correct the typos errors. This will apply to the whole manuscript. Other typos include lack of space before/ after E.g. lines: 23-24: in L(lumbar)1-2, L2-3, L3-4, L4-5, and L5-S(sacrum)1; 74-75: L(lumbar) 1-2, L2-3, L3-4, L4-5, L5-S(sacrum)1; 84: EfficientNet-B5 model[7]; 257: Second, The severity.
It would be interesting to correlate the data in the sections. The information in the introduction and discussion sections needs to be enriched with more recent data and more appropriately correlated.
Answer
Thank you for the valuable and insightful comment. We revised the manuscript according to your suggestions. The revisions are highlighted in yellow in the revised manuscript at the following locations.
line 23-24
line 56-58
line 74-75
line 84
line 184-189
line 201-203
line 216-222
line 236-258
line 275-288
line 291
line 294-296
Round 2
Reviewer 1 Report
The author somewhat addressed my comments.
Reviewer 2 Report
Thank you for all your work!
Reviewer 3 Report
The authors response to all my questions.